# Transcriptomic Dynamics of Active and Inactive States of Rho GTPase MoRho3 in *Magnaporthe oryzae*

**DOI:** 10.3390/jof8101060

**Published:** 2022-10-11

**Authors:** Qian Li, Xi Chen, Lianyu Lin, Lianhu Zhang, Li Wang, Jiandong Bao, Dongmei Zhang

**Affiliations:** 1Meishan Vocational Technical College, Ministerial and Provincial Joint Innovation Centre for Safety Production of Cross-Strait Crops, Fujian Agriculture and Forestry University, Fuzhou 350002, China; 2State Key Laboratory for Ecological Pest Control of Fujian and Taiwan Crops, College of Plant Protection, Fujian Agriculture and Forestry University, Fuzhou 350002, China; 3College of Agronomy, Jiangxi Agricultural University, Nanchang 330045, China

**Keywords:** *Magnaporthe oryzae*, small GTPase, MoRho3, constitutively-active (CA), dominant-negative (DN), RNA-seq

## Abstract

The small Rho GTPase acts as a molecular switch in eukaryotic signal transduction, which plays a critical role in polar cell growth and vesicle trafficking. Previous studies demonstrated that constitutively active (CA) mutant strains, of *MoRho3-CA* were defective in appressorium formation. While dominant-negative (DN) mutant strains *MoRho3-DN* shows defects in polar growth. However, the molecular dynamics of MoRho3-mediated regulatory networks in the pathogenesis of *Magnaporthe oryzae* still needs to be uncovered. Here, we perform comparative transcriptomic profiling of *MoRho3-CA* and *MoRho3-DN* mutant strains using a high-throughput RNA sequencing approach. We find that genetic manipulation of MoRho3 significantly disrupts the expression of 28 homologs of *Saccharomyces cerevisiae* Rho3-interacting proteins, including EXO70, BNI1, and BNI2 in the *MoRho3 CA*, *DN* mutant strains. Functional enrichment analyses of up-regulated DEGs reveal a significant enrichment of genes associated with ribosome biogenesis in the *MoRho3-CA* mutant strain. Down-regulated DEGs in the *MoRho3-CA* mutant strains shows significant enrichment in starch/sucrose metabolism and the ABC transporter pathway. Moreover, analyses of down-regulated DEGs in the in *MoRho3-DN* reveals an over-representation of genes enriched in metabolic pathways. In addition, we observe a significant suppression in the expression levels of secreted proteins suppressed in both *MoRho3-CA* and *DN* mutant strains. Together, our results uncover expression dynamics mediated by two states of the small GTPase MoRho3, demonstrating its crucial roles in regulating the expression of ribosome biogenesis and secreted proteins.

## 1. Introduction

The Rho GTPases, comprising proteins ranging from 21 to 25 kDa, are the largest subfamily of the Ras superfamily of proteins. Rho GTPases functionally regulate the progression of multiple cellular processes, including cell skeleton construction, cell polarization, endocytosis, and exocytosis [1]. Members of the Rho GTPase family showed high sequence similarities. These members also possess a well-conserved core G domain that facilitates the switching (on/off) of the Rho GTPase cascade under different cellular signal transduction regimes [2]. Rho GTPases are switched on when bound to GTP and off when bound to GDP, thus acting as a molecular switch in cellular signal transduction [3]. The active or an inactived states of Rho subfamily proteins are controlled by RhoGAP (GTPase activating protein) and RhoGEF proteins (guanine nucleotide exchange factor), respectively. Specifically, when Rho proteins are in the activated state, they, in turn, activate downstream signals by interacting with corresponding effector proteins. At the same time, GAP accelerates the dehydration of GTP into GDP and thus switches Rho GTPase to inactivate state. On the contrary, GEF proteins catalyze GDP-Rho GTPase binding, resulting in the activation of Rho GTPase. Additionally, GDIs (guanine dissociation inhibitors) inhibit the activity of GEFs by maintaining the GDP-binding state of Rho GTPase [4].

In fungi, Rho GTPases are involved in cell polarization, exocytosis, cellular differentiation, and infection [5,6]. Rho3 was first reported in budding yeast and was demonstrated to be a fungi-specific Rho protein [7,8]. It regulates cell polarization and exocytosis through interactions with Myo2 and Exo70 in fission yeast [8,9]. In *Candida albicans*, a shut-down in the Rho3 promoter resulted in a severe cell polarity defect and a partially depolarized actin cytoskeleton [10]. Deletion of Rho3 results in deficiencies in cell polarity and periodic swelling of hyphal tips in *Candida albicans* and *Ashbya gossypi* [11]. Rho3 may play functions redundantly with Rho4 since knock-out of Rho3 has no phenotypic changes in some fungi, such as *Aspergillus niger* and *Trichoderma reesei* [5,12]. Rho3 is also essential in the plant pathogen *Botrytis cinerea*, in which its loss leads to the suppressed formation of the appressorium and reduced virulence [13]. RNA-seq is a powerful tool for studying changing transcriptional dynamics in different organisms. Previous RNA-seq studies revealed the importance of small GTPases in the development and light response in *Monascus ruber* and *Phycomyces blakesleeanus* [14,15]. Cdc42, another member of the Rho GTP-binding proteins, is demonstrated to be essential for penetration and full virulence in *M. oryzae* [16]. The novel role of Cdc42 in the pyruvate metabolism pathway was uncovered by RNA-seq in the insect-pathogenic fungus *Beauveria bassiana* [17]. While most of the studies were conducted in small GTPase knock-out or knock-down mutants. The transcriptional changes in the constitutively active and dominant-negative states of small GTPases remain elusive.

Rice blast disease, caused by the ascomycete fungus *Magnaporthe oryzae* (Syn. *Pyricularia oryzae*), is one of the most severe diseases of rice. To colonize rice, germ tubes formed by asexual spores from the rice blast fungus differentiate into a specific infection structure, the appressorium, via the polarization of septin and actin components of the cytoskeleton. After the formation of appressoria, a large number of effector proteins are secreted into the plant cells by the following two distinct secretion pathways: conventional or non-conventional protein secretion pathway [18]. Accordingly, effector proteins are divided into apoplastic effectors and cytoplasmic effectors. The former is trafficked by the conventional fungal ER-Golgi secretion pathway, while the latter is secreted by an unconventional secretion pathway and accumulates at the biotrophic interfacial complex in invasive hyphae. Importantly, the loss of fungal exocyst components Exo70 and Sec5 leads to the inefficient secretion of cytoplasmic effectors, demonstrating their essential roles in cytoplasmic effectors’ secretion [18]. Exo70 directly interacts with Rho3, and dominant-negative mutants of Rho proteins abolished their interaction in yeast [9,19]. We previously proposed that the appressorium polarized actin cytoskeleton was affected in the *Morho3* mutant because the appressorium formation is affected in the mutant [20]. Currently, there is no direct evidence pertaining to whether or not MoRho3 exerts a direct influence on the exocytosis pathway and the likely impact on the export of secreted proteins, which consequently affects pathogenesis in *M. oryzae*. Here, we investigated transcriptional profiles associated with Rho3 by sequencing transcriptomes in mutants expressing CA and DN states of MoRho3 protein in *M. oryzae*.

## 2. Materials and Methods

### 2.1. Strain Cultivation and Plant Inoculation

*M. oryzae* mutant strains *MoRho3-CA*, *MoRho3-DN,* and wild-type strain 70-15 have been used in this study. The generation of *MoRho3-CA*/*MoRho3-DN* is described in the previous study [20]. The intact or wounded leaves of 8-day-old barley (*Hordeum vulgare* cv. Gold Promise) and four-week-old rice seedlings (*Oryza sativa* L. cv. TP309) were inoculated with the hyphae suspension at 0.02% Tween solution. The plants were evaluated for disease symptoms at 7 days post inoculation (dpi).

### 2.2. RNA Extraction, Sequencing Library Construction, and Illumina Sequencing

The 10-day-old hyphae samples cultured at 26 °C using liquid complete medium (CM: 0.6% yeast extract, 0.6% casein hydrolysate, 1% sucrose, 1.5% agar) were harvested for RNA extraction. Total RNA was extracted from ground samples with the RNAprep pure Plant Kit (Tiangen, Beijing, China) according to the manufacturer’s instructions. RNA degradation and contamination were checked on 1% agarose gels. RNA integrity was assessed by RNA Nano 6000 Assay Kit of the Agilent Bioanalyzer 2100 system (Agilent Technologies, Santa Clara, CA, USA). In total, 1.5 μg of total RNA per sample was used for the RNA preparations. mRNA was purified from total RNA using poly T oligo attached magnetic beads. Fragmentation of mRNA was performed with divalent cations under elevated temperature in NEBNext First Strand Synthesis Reaction Buffer. Then first-strand cDNA was synthesized using a random hexamer primer. Second strand cDNA was synthesized subsequently with DNA Polymerase I and RNase H. Remaining overhangs were repaired into blunt ends with exonuclease. After adenylation of 3′ ends of DNA fragments, NEBNext Adaptor was ligated. cDNA fragments were purified with AMPure XP system (Beckman Coulter, Beverly, MA, USA). Then PCR was performed with Phusion High Fidelity DNA polymerase. Finally, PCR products were purified (AMPure XP system) and library quality was evaluated on the Agilent Bioanalyzer 2100 system. Sequencing libraries were generated using NEBNext ^®^ Ultra TM RNA Library Prep Kit for Illumina ^®^ (NEB, Ipswich, MA, USA) following the manufacturer’s recommendations. The cDNA library was sequenced on the Illumina sequencing platform (IlluminaHiSeq™ 2000) with 150 bp pair-end (PE) reads length and ~280 bp insert size (Novogene, Beijing, China). An in-house Perl program was used to select clean reads by removing adaptor sequences, low-quality sequences (sQ ≤ 5), and reads with more than 10% N bases.

### 2.3. Reads Alignment and Normalization of Gene Expression Levels

The reference genome for RNA-seq analysis is *M. oryzae* genome 70-15 (GenBank assembly accession GCF_000002495.2) [21]. Clean reads were mapped to reference sequence with Tophat [22,23]. Cufflinks and Cuffmerge were further used to extract all possible exons [24]. Short reads were realigned to all exon sequences by Bowtie2 [25], and expression abundance was calculated by RSEM with default parameters [26]. Low expression genes have been removed. TPM values obtained from RSEM have been used as quantified gene expression levels.

### 2.4. Identification of DEGs and Secreted Proteins

Differential expression analysis was conducted using edgeR [27]. *P* values were subjected to Bonferroni correction test (*q*-value), and a threshold of *q*-value < 0.05 and fold change >2 were used to decide whether significant expression differences exist between samples. Gene expression heatmaps are generated by Genesis [28]. Secreted proteins were annotated in previous study [29], in which proteins with signal peptides and no transmembrane domains were defined as secreted proteins.

### 2.5. Quantitative Real-Time PCR

Gene expression levels obtained from RNA-Seq analysis were validated by quantitative real-time PCR (qRT-PCR) for 18 differentially expressed genes (DEGs, *q*-value < 0.05 and fold change >2). Quantitative SYBR Green qRT-PCR Kit (TaKaRa, Tokyo, Japan) was used to validate the expression changes of DEGs identified by RNA-seq. Relative transcript abundance of genes was calculated by comparing expression level of genes and *beta-tubulin* genes with the 2^−^^∆∆^^Ct^ method. The primers for qRT-PCR used in this study were available in Appendix A.

### 2.6. Functional Annotation and Enrichment Analysis of DEGs

Functional annotation of DEGs was conducted by InterProScan (https://github.com/ebi-pf-team/interproscan, accessed on 15 August 2018) with Pfam database (http://pfam.xfam.org/, accessed on 15 August 2018). ClueGo and CluePedia were used for GO enrichment analysis and a cutoff of adjusted *p*-value < 0.05 were selected for searching enriched GO terms [30,31]. Pathway enrichment analysis of DEGs was performed by KEGG (https://www.genome.jp/eg/, accessed on 15 August 2018). Ribosome biogenesis genes were identified according to GO pathway annotation of *M. oryzae* 70-15 genome.

### 2.7. Identification of Rho3 Interacting Proteins

Protein sequences of yeast Rho3 (ID: YIL118W) interacting proteins were obtained from the Saccharomyces Genome Database (http://www.yeastgenome.org/, accessed on 15 August 2018) [32]. Orthologous proteins of yeast Rho3 interacting proteins were identified by reciprocal BLASTP (Basic Local Alignment Search Tool) searching (E < 1 × 10^−10^) against the *M. oryzae* protein sequences [21,33].

## 3. Results

### 3.1. Transcriptome Sequencing of Constitutively-Active (CA) and Dominant-Negative (DN) Mutants of MoRho3

Phylogenetic analyses suggest that GTPase Rho3 is evolutionary conserved in fungi (Appendix A). We identified five GDP/GTP binding and GTP hydrolysis domains (G1 to G5, Appendix A). CA and DN states are constructed by mutating amino acids of the G1 and G4 domains, respectively (Appendix A). As previously described [20], the CA mutant strains were generated by substituting glycine at position 22 with valine. On the other hand, to generate the DN strains, aspartate at position 128 was substituted with alanine (Figure 1A). The CA and DN sequences were amplified and cloned into the pTE11 plasmid with their expression driven by the RP27 promoter. To understand the transcriptional differences between CA and DN, we conducted RNA sequencing in hyphae samples of a CA mutant, a DN mutant, and the wild-type control 70-15 with 3 biological replicates (Appendix A). We obtained 21–31 million PE150 reads for 9 samples. We mapped the adapter trimmed reads for the 9 samples against the reference genome sequence 70-15 for gene expression calculation and obtained a mapping rate of 81.66% to 84.49% for each sample (Appendix A). Moreover, lowly expressed genes (less than 100 reads in the sum of 9 samples) were filtered out during analyses to enhance the robustness. Principal component analysis (PCA) based on the expression level of genes showed that replicates belonging to different samples were well clustered into three groups, corresponding to WT, CA, and DN strains, respectively (Figure 1B). We further validated the reproducibility of the dataset obtained from RNA-seq by selecting 18 genes that have changes in expression of *MoRho3-CA* vs. WT or *MoRho3-DN* vs. WT with qRT-PCR. The correlation (R2) of expression level alterations (mutant vs. WT) in RNA-seq and qRT-PCR results are ~0.86, demonstrating the high reliability of the sequencing results (Figure 1C and Appendix A).

### 3.2. Comparison of DEGs in MoRho3 CA and DN Mutants

To investigate transcriptional divergence of CA and DN mutants, we set up to obtain DEGs in mutants by comparing RNA-seq data of CA or DN with the wild-type control. A *q*-value < 0.05 and folds-change >2 were used as the analytical thresholds to identify DEGs. In *MoRho3-CA* vs. WT, we detected 874 up-regulated and 1511 down-regulated genes (Figure 2A), while in *MoRho3-DN* vs. WT, we detected 986 up-regulated and 1215 down-regulated genes (Figure 2B,C). Results from comparative DEG analyses revealed a substantial number of genes were down-regulated in the CA and DN mutant strains (Figure 2A,B), suggesting the constant switching on/off of *MoRho3* has a significant impact on gene expression. Of 1050 DEGs, 202/1050 up-regulated and 694/1050 down-regulated genes showed common expression patterns in both *MoRho3-CA and MoRho3-DN* strains (Figure 2D). At the same time, 31/1050 and 123/1050 genes were exclusively up-regulated in the MoRho3-CA and MoRho3-DN, respectively (Figure 2E). From these analyses, we observed that many of the genes were uniquely up/down-regulated in CA or DN mutant strains (DN up = 661, DN down = 490, CA up = 641, CA down = 694). Overall, these results suggest that the mutation of MoRho3 into CA and DN states led to dramatic transcriptional alterations in *M. oryzae*.

### 3.3. Pathways Related to CA and DN Mutation States of MoRho3

To explore the pathways preferentially affected in the two individual mutant strains, we performed pathway enrichment of DEGs with the Gene Ontology (GO) pathway and Kyoto Encyclopedia of Genes and Genomes (KEGG) pathway enrichment methods. Up-regulated genes in *MoRho3-CA* are specifically enriched at pathways of the spliceosome, RNA polymerase, pyrimidine or purine metabolism, TCA cycle, and most significantly enriched at the ribosome biogenesis pathway (Figure 3A). Down-regulated genes recorded in the *MoRho3-CA* strains are enriched in genes associated with different amino acids and chemical metabolism pathways, such as tyrosine, beta-alanine, and starch or steroid metabolisms (Figure 3A). Some metabolic pathways, as well as ABC transporters and regulators for autophagy, are enriched in the upregulated genes of *MoRho3-DN* (Figure 3B). While down-regulated genes identified in the *MoRho3-DN* strains are enriched with genes associated with selected metabolic pathways (Figure 3B). The preference for different pathways in *MoRho3-CA* and *MoRho3-DN* suggests that they execute their functions through different mechanisms (Figure 3C,D).

### 3.4. CA Mutation Enhances the Expression of Ribosome Biogenesis Genes

The ribosome is an essential component of the ribonucleoprotein complex responsible for ribosome biogenesis, in which mRNA is translated into proteins [34]. Recent studies suggest that GTPases are likely to play essential roles in the assembly of ribosomes in bacteria and eukaryotes [35,36]. At the same time, it is still unknown whether Rho GTPase regulates ribosome biogenesis in fungi. However, DEG analyses performed in this study showed a significant number of genes exclusively up-regulated in the *MoRho3-CA* strains enriched in the ribosome biogenesis pathway (Figure 4A,B). To ascertain the influence of MoRho3 on ribosome biogenesis, we further analyzed the expression dynamics of ribosome biogenesis-related genes in the mutant strains. There are 100 ribosome biogenesis genes in the *M. oryzae* genome. The 60/100 are components of the large (60 S) subunit, while the remaining 40/100 belong to the small subunit (40 S). A heatmap showing expression variations indicated that most ribosome biogenesis genes of large or small subunits are up-regulated in *MoRho3-CA* mutant strains, among which 43 and 29 are significantly up-regulated (Figure 4A,B). Conversely, the expression pattern of these genes is relatively unaffected in the *MoRho3-DN* mutant strains. Ribosomal proteins L1 (the largest protein component of the large ribosomal subunit) and L12 exhibited the most significant changes in the *MoRho3-CA* mutant strains.

### 3.5. CA Mutation Regulates Melanin Biosynthesis

To evaluate the impact of CA and DN states on the morphological development of *M. oryzae*, we grew CA and DN mutants in CM medium with WT as a control for 7 days. The colony morphological characterization assay showed that the *MoRho3-CA* grows significantly faster than the WT (Figure 5A,B). More so, we observed that *MoRho3-CA* mutant produced more melanin than WT, as portrayed in the color of the CA colony, which is much darker than the WT colony (Figure 5A). We further investigated pathways that are differentially enriched in up- or down-regulated genes in CA and DN mutants. In addition to ribosome biogenesis genes that are specifically enriched in CA up-regulated genes, we also noticed the enrichment of melanin biosynthesis genes in CA DEGs (Figure 5C). For example, we observed that the expression of MGG_05059, which encodes a scytalone dehydratase and is responsible for full pathogenicity [37], was up-regulated ~7 folds in the CA mutant. The expression of MGG_07216, a 1,3,6,8-tetrahydroxynaphthalene reductase encoding gene, as well as MGG_02252, a 1,3,8-trihydroxynaphthalene reductase encoding gene, was up-regulated a hundred folds in CA mutant (Figure 5D). Therefore, the changes in gene expression of melanin synthesis genes explain the morphological changes observed in the CA mutant strains.

### 3.6. CA and DN Mutations Affect the Expression of Genes Involved in Protein Secretory Pathway

Given that Rho3 is a conserved and fungi-specific GTPase, we hypothesize that the protein-protein interaction network of Rho3 is also conserved, at least partially between budding yeast and *M. oryzae*. To investigate whether Rho3 CA or DN state will affect the expression of the interaction proteins, we first downloaded sequences of Rho3 interacting protein from yeast. We identified orthologous yeast Rho3 interacting proteins in *M. oryzae* through reciprocal best hit of BlastP searching. From this analysis, we obtained 63 putative MoRho3 interacting proteins. By comparing the expression levels of putative MoRho3 interacting proteins, we found that the expression levels of 20 genes were affected in CA or DN with a fold change of >1.5 and 8 genes with a fold change of >2 (Figure 6A). Interestingly, Exo70, a vital component of the exocyst complex [19], is up-regulated exclusively in the *MoRho3-DN* mutant strains. The other components of the exocyst complex, including Sec3, Sec10, and Exo84, were also slightly up-regulated in the *MoRho3-DN* mutant strains. Bub1, a protein involved in the cell cycle [38], is significantly down-regulated in CA and DN mutants. Moreover, we observed a significant up-regulation in the expression of Boi1 and Boi2, which are significantly up-regulated in the CA mutant (Figure 6A). In yeast, Boi1 and Boi2 mediate the fusion of secretory vesicles with the plasma membrane at sites of polarized growth in yeast [39]. To understand how CA and DN states affect the infection capabilities of *M. oryzae*, we inoculated two mutants on intact or wounded rice and barley leaves. We observed that both mutants exhibited impaired pathogenicity on plants, while the decrease in virulence in the CA mutant strains was more severe than that in the DN mutant, particularly on wounded leaves, suggesting CA and DN mutants affect the surface-sensing and penetration stage (Figure 6B). The disrupted expression of exocyst complex genes is in line with the compromised pathogenicity of rice and barley leaves, suggesting that the secretion or expression of effector proteins may also be influenced. To test this, we annotated all secreted proteins in *M. oryzae* and investigated their expression in the mutants. We identified 908 secreted proteins in *M. oryzae*, among which we identified 46 up- and 141 down-regulated in CA, and 89 up- and 99 down-regulated in the DN mutant strains (Figure 6C). Compared with non-secreted proteins, we noticed that the variation of secreted proteins is more significant than non-secreted proteins in both CA and DN mutants, particularly among down-regulated secreted proteins (Figure 6C). Together, we found that the expression dynamics of MoRho3 interacting exocytosis genes is significantly disrupted in CA and DN mutant strains, which may subsequently affect the expression and secretion of effector proteins essential for full pathogenicity.

## 4. Discussion

Blast disease caused by *M. oryzae* is one of the most devastating diseases on food crops such as rice, barley, millet, and most recently, wheat [40,41,42]. The rice blast fungus completes the infection cycle via a specialized cell called an appressorium at the penetration site [43]. Appressorium formation requires sophisticated cytoskeleton organization and cell cycle reprogramming [44,45]. Melanin biosynthesis and turgor generation are essential for appressorium to obtain enough pressure to enter plant cells [46]. The Rho GTPases, which serve as molecular switches, participate in different signaling pathways throughout the appressorium formation process [47,48]. Previous studies revealed that MoRho3 is essential for appressorium formation and pathogenicity [20], while the regulation network underlying these processes is not fully understood. Here, through transcriptomic sequencing, we characterized genome-wide expression changes related to constitutively active and dominant-negative states of MoRho3.

MoRho3, as an evolutionary conserved and fungi-specific small GTPase, plays important roles in polar cell growth, vesicle transport, and pathogenesis [9,20]. We expressed MoRho3 mutated into constitutively active and dominant-negative states in *M. oryzae*. Constitutively active mutations rendered the Rho proteins unable to hydrolyze GTP and maintained an active state. While dominant-negative mutation of Rho proteins makes the protein bind to GEFs, preventing them from activating endogenous Rho and thus maintaining in an inactive state [49]. In general, we observed dramatic transcriptional disruptions in two mutants, particularly in the CA mutant strains, where the number and level of gene downregulation are more obvious than in the DN mutant strains, indicating *MoRho3-CA* may play more important roles in gene regulation.

A previous study showed that the appressorium formation is severely inhibited in the MoRho3 knockout mutant strain, eventually leading to a significant decrease in the ability to infect rice. However, MoRho3 overexpression mutant presented enhanced ability in rice infection [20]. More so, the constitutively active mutant *MoRho3-CA* produces morphologically normal conidia, but the formation of appressorium is defective. The dominant-negative mutant *MoRho3-DN* produces a narrow, elongated conidial, and malformed appressorium. These results suggest that MoRho3 is essential for appressorium formation, whereas two states of MoRho3 may play distinct functions. Indeed, we observed that the two mutants share less than half of the common DEGs, and both possess large groups of unique DEGs. Therefore, we hypothesize that these unique DEGs influence appressorium formation differently.

Through the analysis of the colony growth and color in *MoRho3-CA*/*DN* mutants, we observed that the growth rate of *MoRho3-CA* is faster and produces more melanin than wild type 70-15, which is consistent with the upregulation of melanin biosynthesis genes in *MoRho3-CA*. Although melanin biosynthesis is not affected in *the MoRho3-DN* mutant, there is, however, a significant reduction in the virulence of the *MoRho3-DN* mutant strains, suggesting that additional mechanisms contribute to the full pathogenesis of this *M. oryzae.* Since the generation of melanin is critical for turgor accumulation [46], it may partially explain why the *MoRho3-CA* mutant strains completely lost their virulence.

Transcriptomic analyses identified 2201 and 2385 differentially expressed genes in *MoRho3-CA* and *MoRho3-DN* mutant strains, respectively, among which 1050 DEGs are shared by CA and DN mutants. Functional enrichment analysis of differentially expressed genes revealed that the pathways overrepresented in the down-regulated genes of the two mutants were very similar. However, the pathways overrepresented among the up-regulated genes were quite different. Transcriptional activity, nucleic acid metabolism, and ribosome biogenesis pathways were most significantly affected in *MoRho3-CA* mutants. Ribosomes are ribonucleoprotein complexes responsible for protein biosynthesis in all living organisms [50]. We observed that 72 out of 100 ribosome biogenesis genes were significantly up-regulated in CA mutants, while the expression of ribosome biogenesis-related genes was only slightly affected in DN mutants, which was consistent with the differences in growth rate in the two mutants, indicating that Rho3-CA can directly or indirectly affect ribosomal protein gene expression, thereby affecting cell growth and metabolism. The emerging role of Rho GTPase in regulating ribosomal proteins thus greatly expands our understanding of protein synthesis signal transductions [51].

Meanwhile, Rho3 participates in the non-conventional secretion pathway by the interaction of DN state with Exo70. The observation of Exo70 upregulation in the *MoRho3-DN* mutant is consistent with a previous study showing a dominant-negative mutation of Rho protein abolished the interaction of Rho3 and Exo70 in yeast [8,9], suggesting a specific interaction of Rho3-DN and Exo70. The expression dynamics of predicted secreted proteins revealed that their expression was significantly affected, among which the expression of most secreted proteins was suppressed in CA, probably due to the altered expression of exocytosis genes, which in turn affects the pathogenicity of *M. oryzae*.

It is worth noting that the growth deficiency of MoRho3 CA or DN mutant is more dramatic than its null mutant [20], which is probably because the regulatory roles of a switch on/off states of Rho3 are transient or stage-specific and could be replaced by redundant mechanisms when Rho3 is completely lost. While constant expression of CA or DN states makes the regulatory roles of switch on/off states of Rho3 continuously exist in cellular signaling pathways, it results in profound impacts on phenotypic development. Thus, CA or DN states of Rho proteins driven by stage-specific or inducible promoters will be very informative. In addition, some chemical inhibitors, such as GTP gamma S, can prevent the GTP-binding state of GTPases from being inactivated, which can also be added at different stages to analyze stage-specific functions of GTPases [52]. A recent study in *M. oryzae* with an analog of GTP gamma S demonstrated the important role of PoRal2 in appressorium formation [53].

## 5. Conclusions

Collectively, we reported genome-wide transcriptional dynamics associated with constitutively active and dominant-negative states of MoRho3. We observed dramatic transcriptional changes of Rho3 interacting proteins, such as EXO70, BNI1, and BNI2. We found that ribosome biogenesis genes were significantly up-regulated in the *MoRho3-CA* mutant. Additionally, we observed expression levels of secreted proteins were significantly affected. Through comparative RNA-seq analysis, we defined genes that are related to phenotypic changes, including melanin biosynthesis and pathogenicity of *MoRho3-CA* and *MoRho3-DN*. More functional characterizations are needed to uncover the molecular mechanisms of the genes identified in this study. For example, more efforts are needed to address why mutation of MoRho3 influences secreted proteins and how the disruption of secreted proteins impacts the pathogenicity of *M. oryzae*.

Together, we believe the transcriptomic signatures reported here will shed light on our understanding of the Rho GTPase in filamentous fungi, particularly on pathogenic fungi that threaten human or plant health.

## Figures and Tables

**Figure 1 jof-08-01060-f001:**
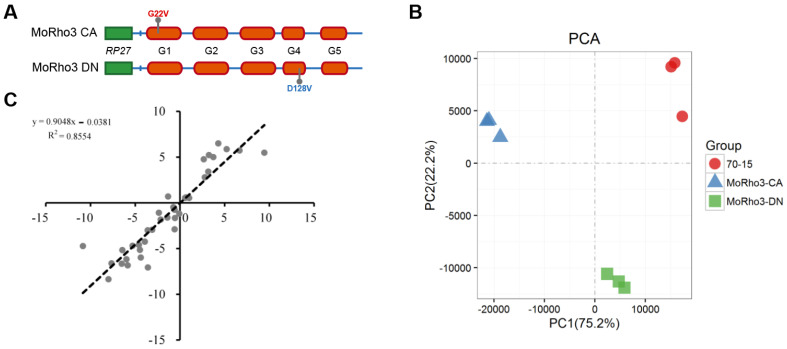
Transcriptome sequencing of Constitutively Active (CA) and Dominant-Negative (DN) mutants of MoRho3. (**A**) Schematic showing generation of CA and DN mutants of MoRho3. (**B**) Principal component analysis of 9 samples based on the expression level of 9230 genes. (**C**) qRT-PCR validation of reproducibility of gene expression level estimated by RNA-seq.

**Figure 2 jof-08-01060-f002:**
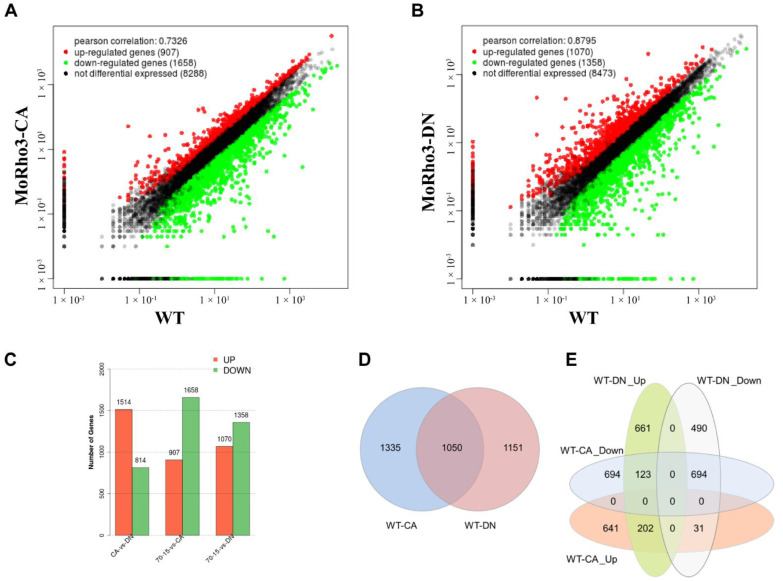
Comparison of DEGs in MoRho3 CA and DN mutants. Expression comparison between *MoRho3-CA* and WT (**A**) and *MoRho3-DN* and WT (**B**), red points represent up-regulated genes, and green points represent down-regulated genes. (**C**) The number of differentially expressed genes identified in *MoRho3-CA* vs. WT and *MoRho3-DN* vs. WT. (**D**) The intersection of all DEGs between CA and DN. (**E**) The intersection of up-regulated and down-regulated genes between CA and DN.

**Figure 3 jof-08-01060-f003:**
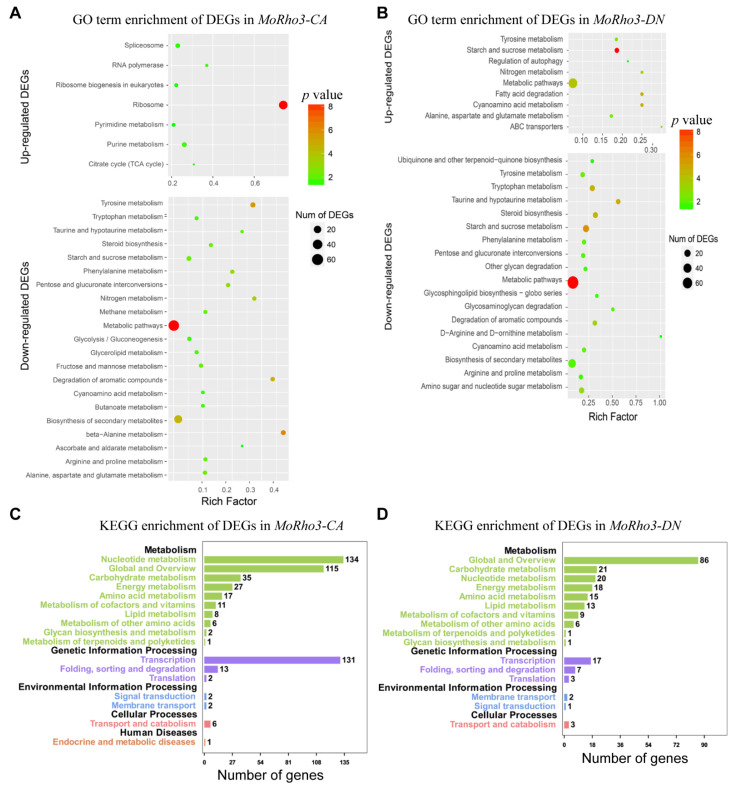
Pathways related to DEGs of CA and DN mutants of MoRho3. (**A**) GO term enrichment of up- and down-regulated genes in *MoRho3-CA*. (**B**) GO term enrichment of up- and down-regulated genes in *MoRho3-DN*. (**C**) Statistics of KEGG pathway enrichment of up- and down-regulated genes in *MoRho3-CA*. (**D**) Statistics of KEGG pathway enrichment of up- and down-regulated genes in *MoRho3-DN*.

**Figure 4 jof-08-01060-f004:**
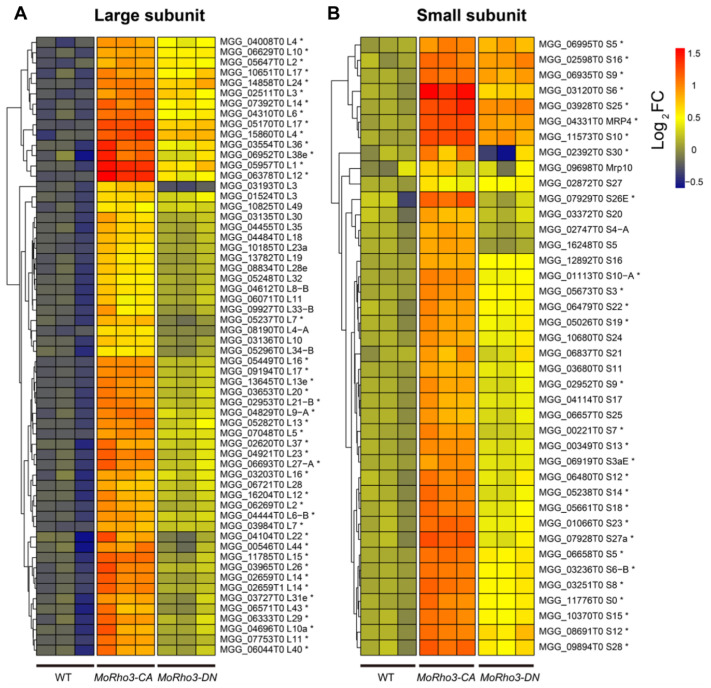
CA mutation enhances the expression of ribosome biogenesis genes. (**A**) Expression heatmap of genes related to ribosomal large subunit. (**B**) Expression heatmap of genes related to ribosomal small (**B**) subunit. * stands for differential expressed gene.

**Figure 5 jof-08-01060-f005:**
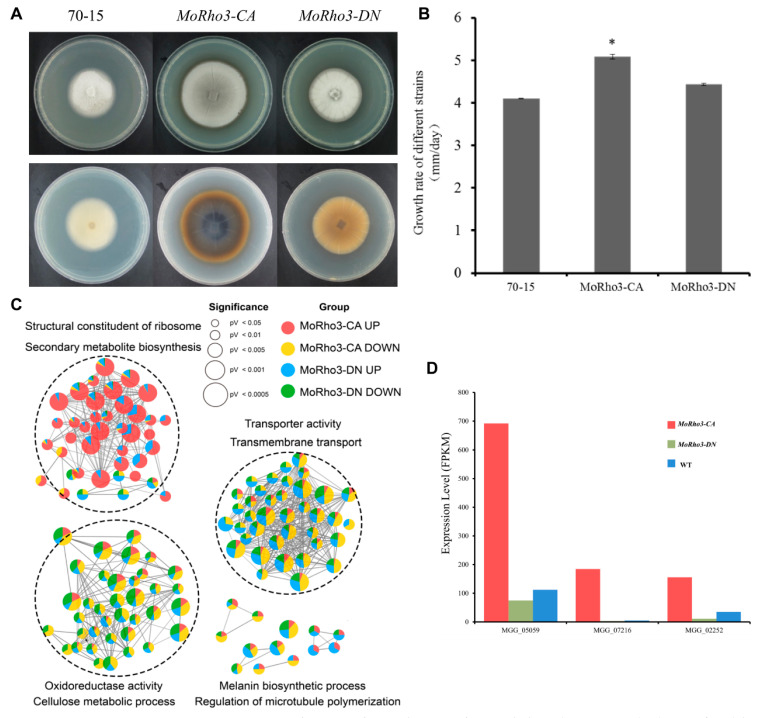
CA mutation regulates melanin biosynthesis. (**A**) Colony morphology of wild-type 70-15, *MoRho3-CA*, and *MoRho3-DN* strains growing on CM medium. (**B**) The growth rate of *MoRho3-CA*, *MoRho3-DN* mutant. Error bar represents standard deviation from three replications and * represent significant difference (*p* < 0.05). (**C**) GO enrichment comparison of up- or down-regulated DEGs between *MoRho3-CA* and *MoRho3-DN* mutant. (**D**) Expression comparison of three melanin biosynthesis genes MGG_05059, MGG_07216, and MGG_02252 in *MoRho3-CA*, and *MoRho3-DN* mutants.

**Figure 6 jof-08-01060-f006:**
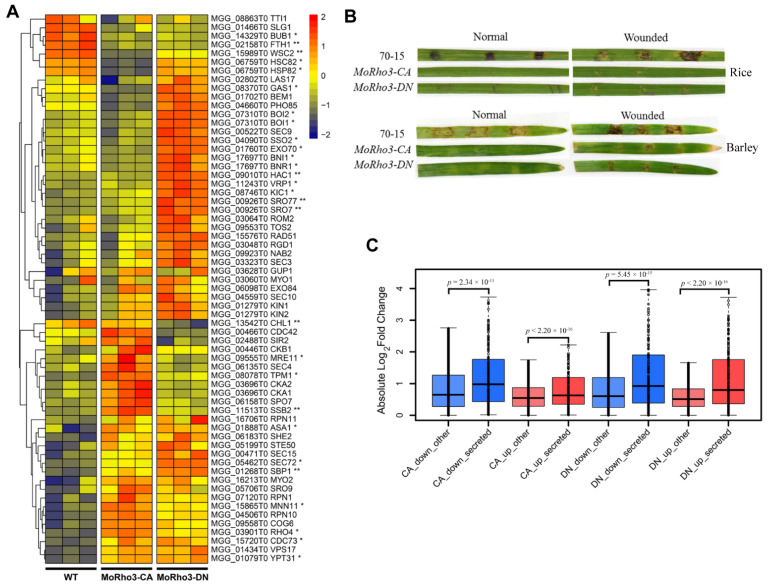
CA and DN mutations affect the expression of genes involved in the protein secretory pathway. (**A**) Expression analysis of orthologous genes of *Saccharomyces cerevisiae* Rho3 interaction proteins in *M. oryzae*. * stands for fold change >1.5. ** stands for fold change >2. (**B**) Pathogenicity assays of rice and barley leaves inoculation with *MoRho3-CA/DN* mutants. (**C**) Comparison of secreted (blue and red) and non-secreted (light blue and light red) protein expression in CA and DN mutants.

## Data Availability

The sequencing reads for this study are available in NCBI under GEO accession number: GSE207559. The authors state that in the current study, the necessary data for the conclusion and Appendix A are included.

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
