# Peer review of "Transcriptomic Dynamics of Active and Inactive States of Rho GTPase MoRho3 in Magnaporthe oryzae"

_jof, 2022, doi:10.3390/jof8101060_

Round 1

Reviewer 1 Report (New Reviewer)

MoRho3, a member of the small GTPase superfamily in the rice blast pathogen, is a virulence factor. Furthermore, loss of function, constitutive-active (CA) and dominant negative (DN) mutants also show developmental phenotypes. Here, a genome-wide expression study characterized the transcriptomic signature of the constitutively-active and dominant-negative states of MoRho3.

The transcriptomic work is of high quality and well-documented.  

Recent reviews (it would be good to add one to the the Introduction) do not seem to mention Rho3 orthologs and their role in secretion pathways and virulence in M. grisea, so the study is a novel contribution to work on signaling in this leading genetic-model pathosystem.

for the Discussion: the concept of CA and DN mutants should be better elaborated. The two "states" exist throughout development, and the pleiotropic phenotypes (Zheng et al. 2007 and this study) agree with this. Surprisingly, in a way, growth is normal even in knockout  mutants (this is also true of many other fungal mutants in signaling cascades). Thus it would be appropriate to point the reader to some ideas for future work. Although the genetic approach is indeed strongest, one could also try transient irreversible activation with GTP-gamma-S to compare phenotypes with those found in the mutants, and perhaps focus to appressorium formation itself. Analogs have been used successfully lfor example in the Pmk1 MAPK pathway of M. oryzae. Returning to the transcriptomics, effects on ribosome biogenesis could again be related to the presence of the CA state throughout development rather than a direct result of Rho3-GTP signaling.

lines 216-219 - confusing, please rewrite. There is no dynamic switching, the mutations are present throughout development. Explain more clearly what can be inferred about gene expression in the wild type, from what was done here, which was to perturb the regulatory network by expression of the CA & DN mutants.

Minor points and edits:

Are Cdc42 and Rho3 orthologs? The reader might not be perfectly familiar with the small GTPase superfamily so it would be good to address this explicitly. Figure S1 would be a perfect place to do this, rather than repeating a phylogeny similar that published previously in ref 19.

Figure 6B goes beyond the result published in ref 19, in that infection of wounded leaves was tested. The result suggests that DA and DN affect the surface-sensing and penetration stage as virulence is increased in wounded leaves. The rescue of virulence is only partial though, suggesting a pleiotropic contribution too. Please discuss.

line 319 - It is fascinating that interacting proteins predicted from yeast are affected in the Rho3 mutants. These were found by BLAST searches, but the details of how the yeast protein interaction maps were mined to extract the interaction network of Rho3 seem to be missing from the methods.

lines 95- 96, remark about the actin cytoskeleton in ref 19 (2007 Eukaryotic Cell paper) - I couldn't find it, please check/reword for example "we proposed previously [19], based on findings in other ascomycetes that, ..." 

Some minor edits found while reading:

line 19 "While.." sentence is incomplete (tracked changes appear in the pdf copy so I might have missed something)

line 48 typo - possed

line 62 - meaning of "most" is unclear, give specific examples or delete; delete "proved to be"

lines 83-4, better: "a specific infection structure, the appressorium, ..."

line 85 "appressorium" replace with "appressoria"

line 105 - mutant strains

line 176 - are switched - this really means "mutant strains were constructed by..."

line 181 - "acid" is erased in the tracked changes, restore or write "aspartate"

 line 262 typo isan

line 267-271 some text is repeated here

line 373 polar cell growth

line 381 - see remark above - CA is not present in the "normal" WT network, please reword

line 387 typo conidia

line 399 typo mutan

line 408 shared by the CA and DN mutants

line 428 "disrupted" better "altered"

line 440 "have influence on" replace "influence"

Author Response

Reviewer 2 Report (New Reviewer)

Extensive editing of English language and style required, preferably, from an expert from a native English speaking country.

Result line 4: change 5 to five

Follow the journal’s format in writing figures.  E.g. Figure 3. Pathways related to DEGs of CA and DN mutants of MoRho3

How did the authors investigate the formation of appresorium in the pathogen by the CA and DN states  of of Rho GTPase MoRho3. Should explain clearly in the text.

Authors reported, “we inoculated two mutants on intact or wounded rice and barley leaves”. Procedure for inoculation was not elaborated. Did you follow Koch’s postutate?

Description of the experiment should improve especially on the gene expression.

Figure 6B. Pathogenicity assays of rice and barley leaves inoculation with MoRho3-CA/DN mutants. What is the difference between the normal and wounded leaves presented? Not clear?

Author Response

This manuscript is a resubmission of an earlier submission. The following is a list of the peer review reports and author responses from that submission.

Round 1

Reviewer 1 Report

In the current study, the authors reported genome-wide transcriptional dynamics associated with constitutively active and dominant-negative states of MoRho3 genome. They found genes related to melanin biosynthesis and pathogenesis in two mutants MoRho3-CA and MoRho3-DN of Magnaporthe oryzae. The data presented in the manuscript is thoughtfully designed, organized, and delivered. The manuscript is well written, and the figures are clearly presented. In addition, this study could add knowledge to the existing knowledge in understating the role of Rho GTPase in fungus and be valuable in the future to control this pathogenic filamentous fungus.

I have only a few minor points for strengthening the manuscript.

The authors could consider rephrasing the title of the manuscript; it is too long and hard to understand at first look.

The authors should consider adding the detail about what samples were used for RNA isolation, how samples were collected and prepared for RNA isolation, and how and using which method RNA was isolated in section 2.2 of materials and methods.

The authors could also add method detail of qRT-PCR in section 2.4 of the materials and methods.  

Consider improving the quality of figures 2C, 3C, 3D, and 5D. For example, color usage in Figures 3C and D is not easy to read and consider changing colors.

 I found the problem in using the article. Therefore, the authors should carefully read and correct the entire manuscript.

Author Response

We thank all reviewers for their positive comments and constructive suggestions. We have revised our manuscript according to the reviewers’ recommendations and provided our point-to-point response below. 

Reviewer #1

In the current study, the authors reported genome-wide transcriptional dynamics associated with constitutively active and dominant-negative states of MoRho3 genome. They found genes related to melanin biosynthesis and pathogenesis in two mutants MoRho3-CA and MoRho3-DN of Magnaporthe oryzae. The data presented in the manuscript is thoughtfully designed, organized, and delivered. The manuscript is well written, and the figures are clearly presented. In addition, this study could add knowledge to the existing knowledge in understating the role of Rho GTPase in fungus and be valuable in the future to control this pathogenic filamentous fungus.

Response:

We thank reviewer for positive comments on our manuscript.

I have only a few minor points for strengthening the manuscript.

The authors could consider rephrasing the title of the manuscript; it is too long and hard to understand at first look.

Response:

We revised the title into “Transcriptomic dynamics associated with constitutively-active and dominant-negative mutants of small GTPase MoRho3 in Magnaporthe oryzae”.

The authors should consider adding the detail about what samples were used for RNA isolation, how samples were collected and prepared for RNA isolation, and how and using which method RNA was isolated in Section 2.2 of materials and methods.

Response:

Thanks for the suggestion. We provided these details in the Section 2.2 (L101-106):

The 10-day-old hyphae samples cultured at 26 °C using liquid complete medium (CM: 0.6% yeast extract, 0.6% casein hydrolysate, 1% sucrose, 1.5% agar) were harvested for RNA ex-traction. Total RNA was extracted from grinded samples with the RNAprep pure Plant Kit (Tiangen, Beijing) according to the manufacturer’s instructions. RNA degradation and contamination was checked on 1% agarose gels. RNA integrity was assessed by RNA Nano 6000 Assay Kit of the Agilent Bioanalyzer 2100 system (Agilent Technologies, CA, USA).

The authors could also add method detail of qRT-PCR in Section 2.4 of the materials and methods.  

Response:

Thanks for the suggestion. We added qRT-PCR in Section 2.5 (L136-143).

2.5 Quantitative real-time PCR

Gene expression levels obtained from RNA-Seq analysis were validated by quantita-tive real-time PCR (qRT-PCR) for 18 differentially expressed genes (DEGs, q-value < 0.05 and fold change > 2). Quantitative SYBR Green qRT-PCR Kit (TaKaRa) was used to vali-date the expression changes of DEGs identified by RNA-seq. Relative transcript abun-dance of genes were calculated by comparing expression level of genes and beta-tubulin gene with the 2-△△Ct method. The primers for qRT-PCR used in this study were available in Table S1.”

Consider improving the quality of figures 2C, 3C, 3D, and 5D. For example, color usage in Figures 3C and D is not easy to read and consider changing colors.

Response:

We apologize for this inconvenience. All figures were compressed when converting from word to pdf in our previous submission. A higher quality figures have been provided in the resubmission.

 I found the problem in using the article. Therefore, the authors should carefully read and correct the entire manuscript.

Response:

We apologize for this inconvenience. All figures were compressed when converting from word to pdf in our previous submission. A higher quality figures have been provided in the resubmission. In addition, the manuscript was edited thoroughly by a native English speaker.

Reviewer 2 Report

The paper describes the effects of two different alleles of rho3 in magnapothe.

One of the analyzed alleles is thought to be domain negative  by inference from other studies. The other is thought to be constitutively active. 

Rho3 is involved in AP formation and hence virulence in Magnaporthe.

Her ethe author report the results of RNA Seq analysis in vegetative Magnaporthe hyphae. 

There two central problems with the paper:

First, it is full of grammatical mistakes that make reading very difficult. In particular the introduction.

Secondly and more importantly, the paper is merely descriptive and provides no insights into the roles of the is particular rho GTPase in the infection process or the biology of Magnaporthe fungi.

Therefore, the manuscript does not meet the criteria required for publication.

Author Response

We thank all reviewers for their positive comments and constructive suggestions. We have revised our manuscript according to the reviewers’ recommendations and provided our point-to-point response below. 

Reviewer #2

The paper describes the effects of two different alleles of rho3 in magnapothe.

One of the analyzed alleles is thought to be domain negative by inference from other studies. The other is thought to be constitutively active. 

Rho3 is involved in AP formation and hence virulence in Magnaporthe.

Her ethe author report the results of RNA Seq analysis in vegetative Magnaporthe hyphae. 

Response:

We thank reviewer for careful reviewing of our manuscript.

There two central problems with the paper:

First, it is full of grammar mistakes that make reading very difficult. In particular the introduction.

Response:

We apologize for the mistakes. The manuscript was edited thoroughly by a native English speaker.

Secondly and more importantly, the paper is merely descriptive and provides no insights into the roles of the is particular rho GTPase in the infection process or the biology of Magnaporthe fungi.

Therefore, the manuscript does not meet the criteria required for publication.

Response:

We have improved our writings to highlight novelty of this paper. In this paper, we reported genome-wide transcriptional dynamics associated with constitutively-active and dominant-negative states of MoRho3. We observed dramatic transcriptional changes of Rho3 interacting proteins. We found that ribosome biogenesis genes were significantly up-regulated in MoRho3-CA mutant. We observed expression levels of secreted proteins were significantly affected. Through comparative RNA-seq analysis, we defined genes that are related to phenotypic changes, including melanin biosynthesis and pathogenicity of MoRho3-CA and MoRho3-DN. RNA-seq is a powerful tool in large-scale biological studies, which provided new insights into small GTPase. We think these contributions are novel for our understanding of Rho3 and its related transcriptional dynamics in pathogenic plant fungus from a different angle.

Reviewer 3 Report

In the current study under review, the authors examine the transcriptional changes associated with states of Rho GTPase MoRho3 in Magnaporthe oryzae. However, some changes need to be made to increase the readability and enhance the presentation of the obtained results. Furthermore, some queries need to be answered/clarified before the manuscript could be judged as suitable for publication. The following are some comments, suggestions, or queries regarding the manuscript.

1.     Introduction is not sufficient. The authors need to mention the previous studies examined the transcriptome of the studied fungus and correlate these studies with their own study highlighting the novelty of their study.

2.     Materials and methods:

a.      How was the consistency of the inoculants determined?

b.     How was the total RNA extracted?

c.      Information regarding the kits used for RNA extraction, cDNA synthesis, etc. needs to be added.

d.     The authors applied TopHat and cufflinks pipeline to analyze their data; however, this pipeline has been improved by the same pipeline developers to produce HISAT2 and StringTie pipeline which according to the developers themselves “accomplish the same tasks while running much faster, using substantially less memory, and providing more accurate overall results.” Please check their manuscript https://doi.org/10.1038%2Fnprot.2016.095. I suggest re-analyzing the data using these tools to have better and more robust results.

e.      What was the criteria to choose the 18 differentially expressed genes for validation by qRT-PCR?

3.     Results:

a.      A full list of expression analysis should be provided as a supplementary material.

b.     Were the raw data of the studied samples deposited on any public database? I recommend doing so.

c.      Figure S1: The full names of the 15 species should be mentioned in the figure’s legend.

d.     The quality of the figures is unacceptable.

e.      The authors mentioned the changes in transcription of genes related to ribosome biogenesis, melanin biosynthesis, and protein secretory pathway; however, such changes were not validated (even for a single gene) via qRT-PCR. I strongly recommend validate the changes in the expression of the most affected genes.

4.     Conclusions in the current manuscript is very short and abbreviated. The authors need to conclude and highlight the main findings of their study, explore the potential obstacles facing their analyses, highlight the future work needed to emphasize their work and build upon it.

Minor comments:

1.     Please mention the full name of the studied organism in the first occurrence.

2.     Several abbreviations were used without explaining them e.g., CA and DN. Please spell out the full term and then use the abbreviation.

Author Response

We thank all reviewers for their positive comments and constructive suggestions. We have revised our manuscript according to the reviewers’ recommendations and provided our point-to-point response below. 

Reviewer #3

In the current study under review, the authors examine the transcriptional changes associated with states of Rho GTPase MoRho3 in Magnaporthe oryzae. However, some changes need to be made to increase the readability and enhance the presentation of the obtained results. Furthermore, some queries need to be answered/clarified before the manuscript could be judged as suitable for publication. The following are some comments, suggestions, or queries regarding the manuscript.

Response:

We thank reviewer for the constructive comments on our manuscript.

  1. Introduction is not sufficient. The authors need to mention the previous studies examined the transcriptome of the studied fungus and correlate these studies with their own study highlighting the novelty of their study.

Response:

Thanks for the suggestion. We added several references which use RNA-seq on fungal small GTPases studies. The novelty of this study was also highlighted by raising points that could be improved in these studies. Specifically, we added the following sentences to Introduction L63-70: “RNA-seq is a powerful tool for studying changing transcriptional dynamics in different organisms. Previous RNA-seq studies revealed the importance of small GTPases in the development and light response in Monascus ruber and Phycomyces blakesleeanus [14,15]. The novel role of Cdc42 in the pyruvate metabolism pathway was uncovered by RNA-seq in the insect-pathogenic fungus Beauveria bassiana [16]. Most of the studies were conducted in small GTPase knock-out or knock-down mutants. The transcriptional changes in con-stitutively-active and dominant-negative states of small GTPase remain elusive.”

  1. Materials and methods:
  2. How was the consistency of the inoculants determined?

Response:

The CA and DN strains were generated in our previous study. More than two candidates were obtained and used for phenotype analyses, qRT-PCR and rice/barley inoculation. In this study, we have confirmed these phenotypes and rice/barley inoculation results of the two strains, which showed consistent results with previous study. We thus think the strains used here is consistent with previous study.

  1. How was the total RNA extracted?

Response:

Thanks for the suggestion. We have provided the following details in the Section 2.2 (L101-107):

The 10-day-old hyphae samples cultured at 26 °C using liquid complete medium (CM: 0.6% yeast extract, 0.6% casein hydrolysate, 1% sucrose, 1.5% agar) were harvested for RNA extraction. Total RNA was extracted from grinded samples with the RNAprep pure Plant Kit (Tiangen, Beijing) according to the manufacturer’s instructions. RNA degradation and contamination was checked on 1% agarose gels. RNA integrity was assessed by RNA Nano 6000 Assay Kit of the Agilent Bioanalyzer 2100 system (Agilent Technologies, CA, USA).

  1. Information regarding the kits used for RNA extraction, cDNA synthesis, etc. needs to be added.

Response:

Thanks for the suggestion. cDNA synthesis information is available at Section 2.2 (L101-107). We have provided the following details about RNA extraction in the Section 2.2:

The 10-day-old hyphae samples cultured at 26 °C using liquid complete medium (CM: 0.6% yeast extract, 0.6% casein hydrolysate, 1% sucrose, 1.5% agar) were harvested for RNA extraction. Total RNA was extracted from grinded samples with the RNAprep pure Plant Kit (Tiangen, Beijing) according to the manufacturer’s instructions. RNA degradation and contamination was checked on 1% agarose gels. RNA integrity was assessed by RNA Nano 6000 Assay Kit of the Agilent Bioanalyzer 2100 system (Agilent Technologies, CA, USA).

  1. The authors applied TopHat and cufflinks pipeline to analyze their data; however, this pipeline has been improved by the same pipeline developers to produce HISAT2 and StringTie pipeline which according to the developers themselves “accomplish the same tasks while running much faster, using substantially less memory, and providing more accurate overall results.” Please check their manuscript https://doi.org/10.1038%2Fnprot.2016.095. I suggest re-analyzing the data using these tools to have better and more robust results.

Response:

TopHat and cufflinks pipeline was used to analyze RNA-seq in this study. We know that upgraded pipelines HISAT2 and StringTie are available. We agree that new pipelines may perform better than old pipelines in running time, memory and overall accuracy. Since we have validated the RNA-seq with qRT-PCR, in which high consistence was observed, we think the main findings from this study will not significantly changed with different pipelines.

  1. What was the criteria to choose the 18 differentially expressed genes for validation by qRT-PCR?

Response:

Gene expression levels obtained from RNA-Seq analysis were validated by quantitative real-time PCR (qRT-PCR) for 18 differentially expressed genes (DEGs, q-value < 0.05 and fold change > 2). To test the robustness of the dataset, we randomly selected these 18 DEGs.

  1. Results:
  2. A full list of expression analysis should be provided as a supplementary material.

Response:

We have uploaded reads file and full list of expression matrix data to GEO under accession number: GSE207559.

  1. Were the raw data of the studied samples deposited on any public database? I recommend doing so.

Response:

Yes, reads file and full list of expression matrix data were deposited on GEO under accession number: GSE207559 before submission. The data will be available upon the acceptance of the paper.

  1. Figure S1: The full names of the 15 species should be mentioned in the figure’s legend.

Response:

Thanks for the suggestion. We added the following sentence in the figure legend:

“Rho3 homologs’ sequences used in the phylogenetic tree are obtained from Trichoderma reesei, Ustilaginoidea virens, Fusarium graminearum, Geaumannomyces graminis, Magnaporthe oryzae, Neurospora tetrasperma, Colletotrichum higginsianum, Blumeria graminis, Botrytis cinerea, Aspergillus nidulans, Saccharomyces cerevisiae, Phytophthora infestans, Puccinia striiformis, Homo sapiens, and Drosophila melanogaster.”

  1. The quality of the figures is unacceptable.

Response:

We apologize for this inconvenience. All figures were compressed when converting from word to pdf. A higher quality figures have been provided in the resubmission.

  1. The authors mentioned the changes in transcription of genes related to ribosome biogenesis, melanin biosynthesis, and protein secretory pathway; however, such changes were not validated (even for a single gene) via qRT-PCR. I strongly recommend validate the changes in the expression of the most affected genes.

Response:

Thanks for the suggestion. RNA-seq results were validated with qRT-PCR. We have randomly selected 18 genes that differentially expressed between mutants and wild type. From this analysis, the correlation (R2) of RNA-seq results and qRT-PCR is 0.8554, suggesting high robustness of the analysis. We thus think it’s not very necessary to select more DEGs to validate the RNA-seq result.

  1. Conclusions in the current manuscript is very short and abbreviated. The authors need to conclude and highlight the main findings of their study, explore the potential obstacles facing their analyses, highlight the future work needed to emphasize their work and build upon it.

Response:

Main findings and potential issues have been fully discussed in Discussion. To highlight main findings and future work of the study, we revised Conclusion (L389-402) into:

“Collectively, we reported genome-wide transcriptional dynamics associated with constitutively-active and dominant-negative states of MoRho3. We observed dramatic transcriptional changes of Rho3 interacting proteins, such as EXO70, BNI1, and BNI2. We found that ribosome biogenesis genes were significantly up-regulated in MoRho3-CA mu-tant. Additionally, we observed expression levels of secreted proteins were significantly affected. Through comparative RNA-seq analysis, we defined genes that are related to phenotypic changes, including melanin biosynthesis and pathogenicity of MoRho3-CA and MoRho3-DN. More functional characterizations are needed to uncover the molecular mechanisms of the genes identified in this study. For example, more efforts are needed to address why mutation of MoRho3 have influence on secreted proteins and how the dis-ruption of secreted proteins impact pathogenicity of M. oryzae. Together, we believe the transcriptomic signatures reported here will shed light on our understanding of the Rho GTPase in filamentous fungi, particularly on pathogenic fungi that threaten human or plant health.

Thanks for the suggestion.

Minor comments:

  1. Please mention the full name of the studied organism in the first occurrence.

Response:

Thanks for the suggestion. Full name has been provided in the first occurrence at Line 20 and Line 71.

  1. Several abbreviations were used without explaining them e.g., CA and DN. Please spell out the full term and then use the abbreviation.

Response:

Thanks for the suggestion. We have thoroughly revised the manuscript so that full name of all abbreviations is given.